# Integration of Transcriptome and Metabolome Reveals Wax Serves a Key Role in Preventing Leaf Water Loss in Goji (*Lycium barbarum*)

**DOI:** 10.3390/ijms252010939

**Published:** 2024-10-11

**Authors:** Xingbin Wang, Sitian Li, Xiao Zhang, Jing Wang, Tong Hou, Jing He, Jie Li

**Affiliations:** 1College of Forestry, Gansu Agricultural University, Lanzhou 730070, China; 17793473313@163.com (X.W.); zx731205@163.com (X.Z.); ing1130652421@163.com (J.W.); ht212119@163.com (T.H.); hejing268@aliyun.com (J.H.); 2College of Horticulture, Gansu Agricultural University, Lanzhou 730070, China; lisitian121@163.com; 3Wolfberry Harmless Cultivation Engineering Research Center of Gansu Province, Lanzhou 730070, China

**Keywords:** wax synthesis, alkanes, alcohols, differentially expressed genes (DEGs), transcription factors (TFs)

## Abstract

Drought stress is one of the main abiotic stresses that limit plant growth and affect fruit quality and yield. Plants primarily lose water through leaf transpiration, and wax effectively reduces the rate of water loss from the leaves. However, the relationship between water loss and the wax formation mechanism in goji (*Lycium barbarum*) leaves remains unclear. ‘Ningqi I’ goji and ‘Huangguo’ goji are two common varieties. In this study, ‘Ningqi I’ goji and ‘Huangguo’ goji were used as samples of leaf material to detect the differences in the water loss rate, chlorophyll leaching rate, wax phenotype, wax content, and components of the two materials. The differences in wax-synthesis-related pathways were analyzed using the transcriptome and metabolome methods, and the correlation among the wax components, wax synthesis genes, and transcription factors was analyzed. The results show that the leaf permeability of ‘Ningqi I’ goji was significantly lower than that of ‘Huangguo’ goji. The total wax content of the ‘Ningqi I’ goji leaves was 2.32 times that of the ‘Huangguo’ goji leaves, and the epidermal wax membrane was dense. The main components of the wax of ‘Ningqi I’ goji were alkanes, alcohols, esters, and fatty acids, the amounts of which were 191.65%, 153.01%, 6.09%, and 9.56% higher than those of ‘Huangguo’ goji, respectively. In the transcriptome analysis, twenty-two differentially expressed genes (DEGs) and six transcription factors (TFs) were screened for wax synthesis; during the metabolomics analysis, 11 differential metabolites were screened, which were dominated by lipids, some of which, like D-Glucaro-1, 4-Lactone, phosphatidic acid (PA), and phosphatidylcholine (PE), serve as prerequisites for wax synthesis, and were significantly positively correlated with wax components such as alkanes by the correlation analysis. A combined omics analysis showed that DEGs such as *LbaWSD1*, *LbaKCS1*, and *LbaFAR2*, and transcription factors such as *LbaMYB306*, *LbaMYB60*, and *LbaMYBS3* were strongly correlated with wax components such as alkanes and alcohols. The high expression of DEGs and transcription factors is an important reason for the high wax content in the leaf epidermis of ‘Ningqi I’ goji plants. Therefore, by regulating the expression of wax-synthesis-related genes, the accumulation of leaf epidermal wax can be promoted, and the epidermal permeability of goji leaves can be weakened, thereby reducing the water loss rate of goji leaves. The research results can lay a foundation for cultivating drought-tolerant goji varieties.

## 1. Introduction

Drought stress affects the normal growth and development of plants to a great extent [1,2]. The structure and chemical composition of plant cuticular wax helps plants to resist various biotic and abiotic invasions [3]. This wax is insoluble in water and soluble in organic solvents. It covers the surface of various plant organs and tissues, and plays an important role in plant adaptation to drought environments [2], in resistance to various pests and diseases [4,5,6], as well as ultraviolet radiation [7], and in limiting non-stomatal water loss in plants [8,9].

Plants with a high wax content exhibit greater resistance to drought stress than plants with a low wax content [10]. The wax of the plant epidermis is regulated by multiple genes and transcription factors. For example, it was found that the overexpression of *OsFAR1* and *MsKCS10* promoted the accumulation of wax in the leaf epidermis of rice (*Oryza sativa*) and tobacco (*Nicotiana tabacum*), in order to enhance drought resistance [11,12]. *OsLKP2*–*OsGI* interaction inhibits the accumulation of wax on the leaf surface, thereby reducing the resistance of rice (*Oryza sativa*) to drought stress [13]. The expression levels of *CsKCS3* and *CsKCS18* in tea (*Camellia sinensis*) are correlated with the wax content in their leaves [14]. *BdCER1-8* plays a dominant role in VLC alkane biosynthesis in *Brachypodium distachyon* leaves [15]. Additionally, transcription factors such as MYB94, MYB96, and MYB30 can activate the wax synthesis process of a plant’s epidermis and promote the accumulation of wax in the epidermis [16,17,18]. Different wax components may affect plant drought resistance differently: the synthesis of primary alcohols can be regulated by *TaFAR2*, *TaFAR3*, and *TaFAR4* to promote the wax content of the leaf epidermis, which ultimately improves drought tolerance in wheat (*Triticum aestivum*) [19]; the overexpression of *AtCER1* in *Arabidopsis thaliana* promotes the biosynthesis of long-chain alkanes, thereby reducing the permeability of the leaf epidermis and improving its drought resistance [20]. Despite the extensive research on wax synthesis, regulation, and function in other plants, our understanding of wax synthesis, regulation, and drought resistance in goji leaves is still scarce.

Goji (*Lycium barbarum*) is a perennial shrub belonging to the family of Solanaceae. Flavonoids, polysaccharides, and carotenoids, as well as other nutrients, are abundant in its fruit [21,22]. Known as a “super food”, it is a famous medicine and food homology plant. Goji planting plays an essential role in the rural economy of Northwestern China, where drought is a vital factor that affects the growth and fruit quality of goji [23,24]. However, information regarding the synthesis of goji leaf wax and the relationship between the wax and drought resistance of goji are still unknown. Therefore, it is of great significance to study wax in different varieties of goji leaves and its mechanism for inhibiting excessive water loss. In this study, ‘Ningqi I’ goji (*Lycium barbarum* cv. ‘Ningqi I’) and ‘Huangguo’ goji (*Lycium barbarum* var. auranticarpum K. F. Ching), which have significant differences in their leaf wax contents, were used to determine the differences in leaf water loss rate and chlorophyll leaching rate between the two kinds of goji. The differences in the morphology and chemical composition of the leaf wax were compared, and the wax metabolism pathway and biosynthesis-related genes in the leaves were analyzed. The aim of this study was to provide a theoretical foundation for understanding the drought resistance mechanisms of goji, for the potential development of drought-tolerant goji varieties.

## 2. Results

### 2.1. Comparison of Leaf Phenotype and Epidermal Permeability between ‘Ningqi I’ Goji and ‘Huangguo’ Goji

The leaf color of ‘Ningqi I’ goji is darker than that of ‘Huangguo’. The ‘Ningqi I’ goji has smooth skin with an obvious luster (Figure 1A), whereas the leaf epidermis of ‘Huangguo’ goji is rough without an obvious luster (Figure 1B). The water loss rate determination experiment showed that the water loss rate of the ‘Huangguo’ goji leaves was significantly higher than that of the ‘Ningqi I’ goji leaves (Figure 1C). Therefore, after 36 h of leaf dehydration, the leaves of ‘Huangguo’ goji were more curled than those of ‘Ningqi I’ goji (Figure 1B). In addition, the determination of the chlorophyll leaching rate showed that the chlorophyll leaching rate of the ‘Huangguo’ goji leaves was much higher than that of the ‘Ningqi I’ goji leaves (Figure 1D). These results show that the epidermal permeability of the ‘Huangguo’ goji leaves was significantly higher than that of the ‘Ningqi I’ goji leaves.

### 2.2. Observation of Wax Structure and GC-MS Analysis of Wax Components in Leaves of ‘Ningqi I’ Goji and ‘Huangguo’ Goji

The wax accumulation in the leaf epidermis of ‘Ningqi I’ goji was dense, showing a concentrated distribution trend, and the wax was arranged closely (Figure 2). The leaf surface was covered with a wax film (Figure 2A,B). On the contrary, the wax accumulation in the leaf epidermis of ‘Huangguo’ goji was not obvious, showing sporadic distribution, and no complete wax film was formed (Figure 2C,D). In addition, the cuticular wax morphology of the two kinds of goji leaves was also different. The cuticular wax crystal morphology of ‘Ningqi I’ goji leaves was rod-shaped, while that of Huangguo’ goji leaves was flaky.

The total wax content of the ‘Ningqi I’ goji leaves was 19.31 μg/cm^2^, which is 2.32 times that of ‘Huangguo’ goji (8.31 μg/cm^2^) (Figure 3A). The cuticular wax of the ‘Ningqi I’ goji and ‘Huangguo’ goji leaves was mainly composed of alkanes, esters, fatty acids, alcohols, and aldehydes (Figure 3B,C). The proportions of alkanes, esters, fatty acids, alcohols, and aldehydes in the leaf epidermis wax of the ‘Ningqi I’ goji were 82.59%, 5.45%, 4.61%, 4.18%, and 0.08%, respectively (Figure 3B). The proportions of alkanes, esters, fatty acids, alcohols, and aldehydes in the cuticular wax of ‘Huangguo’ goji leaves were 65.85%, 11.95%, 11.72%, 3.84%, and 0.47%, respectively (Figure 3C). The proportions of alkanes, esters, alcohols, and aldehydes in the cuticular wax of ‘Ningqi I’ goji were significantly higher than those of ‘Huangguo’ goji. However, the proportion of fatty acids of ‘Ningqi I’ goji was lower than that of ‘Huangguo’ goji (Figure 3B,C).

According to the length of the carbon chain, the alkanes in the cuticular wax of two kinds of goji leaves were analyzed. The results show that the alkanes in the cuticular wax of the two types of goji leaves were mainly alkanes with carbon chain lengths of 31 and 33 (Figure 3D,E). The contents of the two substances in ‘Huangguo’ goji were significantly lower than those in ‘Ningqi I’ goji. The alkane analysis of 31 carbon chain lengths showed that the substance was undecane; the alkane analysis of 33 carbon chain lengths showed that the substance was tritriacontane.

Based on the length of the carbon chains, the alcohol compounds in the cuticular wax of two kinds of goji leaves were analyzed. The results show that they were mainly alcohols with carbon chain lengths of 16, which were significantly less abundant in ‘Huangguo’ goji than in ‘Ningqi I’ goji. The analysis of this alcohol compound with a carbon chain length of 16 showed that the substance was phytol (Figure 3F).

Again, based on the length of their carbon chains, the fatty acids in the cuticular wax of two kinds of goji leaves were analyzed. The results show that they were mainly fatty acids with a carbon chain length of 19 (Figure 3G), which were more abundant in ‘Huangguo’ goji leaves than in ‘Ningqi I’ goji leaves. The analysis of this fatty acid with a carbon chain length of 19 showed that the substance was nonadecanoic acid. In conclusion, the GC-MS analysis of the wax components in the leaves showed that the wax components with significant differences in abundance between the two types of goji leaves were tricosane, tritriacontane, phytol, and nonadecanoic acid.

### 2.3. Metabolomics Analysis of ‘Ningqi I’ Goji and ‘Huangguo’ Goji Leaves

In order to better understand the differences in metabolites between ‘Ningqi I’ goji leaves and ‘Huangguo’ goji leaves, UPLC-MS/MS was used to analyze their metabolic characteristics. A total of 5552 metabolites were detected in the positive and negative ion modes, which were divided into 10 categories (Appendix A) and evaluated using principal component analysis (PCA) (Appendix A). The quality control (QC) sample is located at the center of the score plot, indicating that the instrument is stable and that the results are highly repeatable and reliable. The two principal components (PC1 and PC2) accounted for 79.48% of the total variance. The PCA score plot showed a high clustering of metabolites in the leaves of the two goji species, indicating a significant difference in the metabolite profiles of the samples.

According to the combination of VIP > 1 and *p*-value < 0.05, a total of 829 differential metabolites were further screened (Figure 4A). Among them, 335 differential metabolites were up-regulated in ‘Ningqi I’ goji, compared with ‘Huangguo’ goji. Of the differential metabolites, 494 were relatively down-regulated. The top fifty metabolites with significant differences were classified, including ten glycosides, nine lipids, seven flavonoids, six polysaccharides, five amino acids, four other compounds, three hormone-related substances, two ketones, two lactones, one saponin, and one alkaloid (Figure 4B). The cluster analysis showed that the metabolites were different between the two goji samples (Figure 4C).

The identified different metabolites were classified according to their type, and the results are shown in Figure 4B. The results show that glycosides, lipids, flavonoids, and lactones were the main different metabolites. Eight lipids, two lactones, and one flavonoid compound were significantly up-regulated in ‘Ningqi I’ goji leaves compared with ‘Huangguo’ goji leaves. Free fatty acids, phosphatidic acid (PA), phosphatidylcholine (PE), phosphatidylinositol (PI), and diacylglycerol (DG) were identified as different lipid metabolites. Among them, fatty acids, phosphatidic acid (PA), and other lipids were significantly down-regulated in ‘Huangguo’ goji leaves, compared with ‘Ningqi I’ goji leaves, which was consistent with the trend in wax content difference. Dehydroascorbate (bicyclic form) and D-Glucaro-1,4-Lactone were the main differential metabolites that cause differences in wax content. A significantly up-regulated metabolite (6-Hydroxykaempferol 3-Rutinoside) was identified in flavonoids, which was consistent with the difference in wax content observed. Further analysis of the relative content of these differential components showed that the relative content changes in fatty acids, phosphatidic acid (PA), phosphatidylcholine (PE), inositol phosphate (PI), diacylglycerol (DG), lactones, and flavonoids were the same as the trend in wax content. These differential components may be the reason for the different wax content of the two species of goji leaves. Its specific constituents include 6-O-(13-Methyl-Myristoyl)-6′-O-13-Methyl-Myristoyltrehalose, PA(20:3(8Z,11Z,14Z)-2OH(5,6)/22:4(7Z,10Z,13Z,16Z)), Linolipin A, DG(i-20:0/0:0/LTE4), PE(20:1(11Z)/19:1(9Z)), 1,3-(8r,9r-Epoxy-Octadec-13z,15z-Dien-4,6-Diynoyl)-2-(Myristoyl)-Sn-Glycerol, C16 Sphinganine, PI(22:6(4Z,7Z,11E,13Z,15E,19Z)-2OH(10S,17)/20:1(11Z)), dehydroascorbate (bicyclic form), D-Glucaro-1, 4-Lactone, and 6-Hydroxykaempferol 3-Rutinoside (Appendix A). These substances may play a significant role in the synthesis of wax.

### 2.4. Correlation Analysis between Dominant Components of Wax and Differential Metabolites

In order to further understand the relationship between the dominant components of wax and differential metabolites, a correlation analysis was carried out in this study (Figure 5). The results show that there was a significant correlation between the dominant components of wax and differential metabolites. Nonadecanoic acid was significantly correlated with D-Glucaro-1,4-Lactone and dehydroascorbate (bicyclic form) (*p* ≤ 0.01), while untriacontane, tritriacontane, phytol, and nonadecanoic acid were also significantly positively correlated with differential metabolites such as phosphatidic acid (PA) and phosphatidylcholine (PE) (*p* ≤ 0.05). The correlation coefficients of 6-Hydroxykaempferol with untriacontane and tritriacontane were 0.97 and 0.96, respectively. The correlation coefficients of DG (i-20: 0/0: 0/LTE4) with tritriacontane, phytol, and undecane were 0.97, 0.96, and 0.95, respectively. Therefore, this study suggests that differential metabolites such as phosphatidylcholine (PE), 6-Hydroxykaempferol, and DG (i-20: 0/0: 0/LTE4) are involved in the synthesis of wax components such as alkanes, alcohols, and fatty acids in the leaf epidermis of goji, which in turn makes the total amount of wax in the epidermis of ‘Ningqi I’ goji leaves much higher than in ‘Huangguo’ goji leaves.

### 2.5. Screening of Wax Synthesis-Related Genes

In order to explore the molecular mechanism of wax composition and the differences in abundance between ‘Ningqi I’ goji leaves and ‘Huangguo’ goji leaves, we previously performed transcriptome sequencing and analysis on the leaves of ‘Ningqi I’ goji and ‘Huangguo’ goji [25] (https://ngdc.cncb.ac.cn/gsa, accessed on 11 March 2024). Further transcriptome analysis of the wax synthesis pathway showed that a total of twenty-two DEGs were detected in the complete wax synthesis pathway, including the fatty acid de novo synthesis pathway, the ultra-long-chain fatty acid synthesis pathway, the acyl reduction pathway, and the deacylation pathway, in which there were four, three, eight, and seven DEGs detected, respectively (Figure 6). The expression levels of *LbaACC* (Lba06g02066), *LbaFATB* (Lba05g01237), Lba05g02078, and the long-chain acyl-CoA synthase 2 (LACS2)-encoding gene, Lba06g03261, in the ‘Ningqi I’ goji leaves were 2.36–160.28 times higher than those in the ‘Huangguo’ goji leaves, respectively. In addition, *LbaFATB* (Lba05g02078) was only expressed in ‘Ningqi I’ goji leaves.

The β-ketoacyl-CoA synthase (KCS) family is involved in catalyzing the extension of VLCFA. Three KCS transcripts were differentially expressed, namely *LbaKCS1* (Lba01g00543), *LbaKCS11* (Lba04g01329), and *LbaKCS20* (Lba05g02485). The expression levels of *LbaKCS1*, *LbaKCS11*, and *LbaKCS20* in the ‘Ningqi I’ goji leaves were 2.10–30.05 times higher than those in ‘Huangguo’ goji leaves.

The synthesized VLCFA can be used as a substrate to participate in the decarbonylation pathway and acyl reduction pathway in order to produce aldehydes, alkanes, secondary alcohols, primary alcohols, ketones, and esters. The analysis of DEGs in the decarboxylation pathway showed that the expression levels of genes (Lba06g02387 and Lba01g00793) in the ultra-long-chain aldehyde decarbonylase 1 (CER1) and the gene Lba01g01429 in the alkane hydroxylase (MAH1) family were down-regulated in ‘Ningqi I’ goji leaves. The expression levels of Lba03g02612, Lba08g01893, Lba10g01984, Lba10g01985, and Lba10g01990 were significantly up-regulated in ‘Ningqi I’ goji. In the acyl reduction pathway, the expression level of fatty acyl coenzyme A reductase 2 (FAR2) (Lba01g00847) in the ‘Ningqi I’ goji leaves was lower than that in the ‘Huangguo’ goji leaves; the expression levels of fatty acyl-CoA reductase 2 (FAR2) (Lba11g02102) and wax ester synthase (WSD1) (Lba05g01733, Lba05g01728, Lba06g01907, Lba06g01910, Lba09g00988, and Lba05g01729) in ‘Ningqi I’ goji leaves were significantly higher than those in the ‘Huangguo’ goji leaves.

### 2.6. Screening of Transcription Factors Related to Wax Synthesis

TFs play an important role in regulating genes that are involved in wax biosynthesis by setting the threshold to a q-value of 1. In the transcriptome data, 19 WRKY and 42 MYB transcription factors related to wax biosynthesis were screened and their FPKM expression patterns were analyzed, as shown in Figure 7. By setting the threshold FPKM ≥ 1, six positively regulated transcription factors were selected, which were *LbaMYBS3*, *LbaMYB48*, *LbaMYB3*, *LbaMYB306*, *LbaMYB60*, and *LbaWRKY68*. Their average expression levels in ‘Ningqi I’ goji leaves were over twice as high as those in ‘Huangguo’ goji leaves. Further RT-qPCR analysis of these six genes showed that the changes in expression levels of the six genes were consistent with the results of transcriptome data. The correlation between these six transcription factors controlling wax synthesis, wax components, and total wax content was analyzed. The results show that the expression levels of six wax synthesis-related transcription factors were positively correlated with the total wax content. The RT-qPCR results show that the expression levels of *LbaMYB3*, *LbaMYB48*, and *LbaMYB306* were significantly higher than those of the control. Among them, *LbaMYB3* had the highest expression level.

### 2.7. The Expression of DEGs and TFs in Wax Biosynthesis Was Analyzed Using RT-qPCR

In order to verify the accuracy of the transcriptome data, we randomly selected two key DEGs (*LbaCER1* (Lba06g02387, Lba01g00793)) and six TFs (*LbaMYBS3*, *LbaMYB48*, *LbaMYB3*, *LbaMYB306*, *LbaMYB60*, and *LbaWRKY68*) related to wax synthesis for real-time quantitative PCR (Figure 8A–H). The results show that, except for *LbaCER1* (Lba06g02387) and *LbaCER1* (Lba01g00793) (Figure 8A,B), the expression levels of the other six TFs were consistent with the transcriptome data (Figure 8C–H), which proved that the transcriptome data were reliable. The expression levels of two DEGs and six TFs in the ‘Ningqi I’ goji leaves were significantly higher than those in the ‘Huangguo’ goji leaves; these results were consistent with the trend in wax content differences in the leaf epidermis of the two species of goji, and further confirmed that these DEGs and TFs may be involved in wax synthesis for the leaf epidermis of gojis. It is worth noting that the expression level of *LbaMYB3* was the highest, and was 6.2 times higher in the ‘Ningqi I’ goji leaves than in the ‘Huangguo’ goji leaves (Figure 8C). This suggests that *LbaMYB3* may be one of the main transcription factors regulating wax synthesis in goji leaves.

### 2.8. Combined Metabolome and Transcriptome Analysis

In order to further clarify the molecular mechanism of wax synthesis in goji leaf epidermis, we conducted a combined analysis of metabolome and transcriptome data. As shown in Figure 9, differential genes and differential metabolites were mapped to several pathways. The results show that the pathways related to wax synthesis mainly included fatty acid degradation, terpenoid backbone biosynthesis, ABC transporters, cutin, suberine, and wax biosynthesis, linoleic acid metabolism, alpha-linolenic acid metabolism, and flavonoid biosynthesis. Among them, the metabolites in the cutin, suberine, and wax biosynthesis pathway directly related to wax synthesis were significantly enriched (*p* ≤ 0.05), and the DEGs in the fatty acid degradation pathway were significantly enriched (*p* ≤ 0.01).

In order to understand the relationship between the differential genes of metabolomics and transcriptomics and the pathways of differential metabolites, the interaction between the two omics was studied more systematically. We used KGML to plot the links between these pathways into a network diagram, and obtained a total of nine functional network clusters (Figure 10), including amino acid metabolism, the biosynthesis of other secondary metabolites, carbohydrate metabolism, lipid metabolism, the metabolism of cofactors and vitamins, metabolism of other amino acids, the metabolism of terpenoids and polyketides, nucleotide metabolism, and translation processes. Most of the functional network groups are well connected with enzymes and metabolites involved in wax biosynthesis and its regulation.

According to the FPKM value, seven candidate genes and six transcription factors with the highest expression level were screened out. The correlation among these transcription factors and DEGs and six wax components was analyzed (Figure 11). The results show that the expression levels of *LbaLACS2* (Lba06g03261), *LbaWSD1* (Lba05g01733), *LbaWSD1* (Lba05g01729), *LbaMYBS3* (Lba07g01201), and *LbaMYB306* (Lba07g00856) were significantly positively correlated with the total wax content and alkane and alcohol content in goji leaves (*p* ≤ 0.01). The correlation coefficient of *LbaWSD1* (Lba05g01733) and *LbaMYBS3* expression with total wax content and alkane content in goji leaves was 0.99. The expression level of *LbaMYB60* (Lba08g01828) was significantly positively correlated with the alcohol contents (*p* ≤ 0.01); the correlation coefficient was 0.94. The expression levels of *LbaMYB60* (Lba08g01828) and *LbaWRKY68* (Lba06g01328) were significantly positively correlated with the total wax content and alkane content in goji leaves (*p* ≤ 0.05). The correlation coefficients of *LbaMYB60* expression with total wax content and alkane content in leaves were 0.83 and 0.82, respectively. The correlation coefficient of *LbaWRKY68* expression with total wax content and alkane content in the leaves was 0.85. In addition, the selected genes and transcription factors had a certain but not significant correlation with fatty acid content, ester content, and other unidentified components.

## 3. Discussion

Plant epidermal wax, as the first barrier against external stressors, can limit the non-stomatal diffusion of water from plants and protect plants from drought stress [2,8]. Our study showed that, compared with ‘Huangguo’ goji leaves, the color of ‘Ningqi I’ goji leaves was darker (dark green) and their surfaces were glossy and grayish white (Figure 1A,B). In addition, our chlorophyll leaching experiment and water loss test showed that the epidermal permeability of ‘Ningqi I’ goji leaves was significantly lower than that of ‘Huangguo’ goji leaves (Figure 1C,D) (following a darkness treatment to close the stomata before measurements were taken). These results indicate that the non-stomatal water loss in the leaves of ‘Ningqi I’ goji is reduced due to the lower permeability of the leaf epidermis. It has been previously reported that the permeability of plant epidermis is mainly determined by cuticular wax [26]. Therefore, we further analyzed the wax of the two types of goji leaf.

This further analysis showed that the total wax content of ‘Ningqi I’ goji leaves was significantly higher than that of ‘Huangguo’ goji leaves (Figure 3A). The total wax content of the ‘Ningqi I’ goji leaves was 19.31 μg/cm^2^ (Figure 3A), which was then compared with that of the model plants *Arabidopsis thaliana* and ‘Newhall’ navel orange (*Citrus sinensis* Osbeck cv. Newhall); the ‘Ningqi I’ goji leaves’ total wax content was about fourteen times that of Arabidopsis leaves [27], and about three times that of Newhall leaves [28]. These results suggest that wolfberry, as a shrub adapted to arid environments, has higher levels of wax accumulation than plants that thrive in mild environments, providing important support for environmental adaptation. Therefore, we speculated that the accumulation of wax in the leaf epidermis of goji might reduce the permeability of goji epidermis and thus reduce water loss. It has previously been shown that the accumulation of wax in the epidermis of tomato leaves reduced the permeability of the leaf and improved the drought resistance of tomato [29]. This is basically consistent with our research results. Most of the outer layer of the leaf was formed by epidermal wax, which accumulated in the plant epidermis in the form of a microcrystalline structure [30]. These leaf wax crystalline forms can be categorized into 23 distinct types, including flake, tubular, and filamentous types [30]. The wax crystal morphology of the epidermis of ‘Ningqi I’ goji leaves was flaky and rod-shaped (Figure 2A,B), and the wax crystal morphology of the epidermis of ‘Huangguo’ goji leaves was flaky (Figure 2C,D). The difference lies in the epidermal wax morphology of goji leaves. This may be affected by wax content and composition, so we further analyzed the epidermal wax composition.

Wax components are the key factors affecting the permeability of plant epidermis [31]. Plant cuticular wax mainly includes aliphatic (alkanes, olefins, aldehydes, alcohols, ketones, and terpenoids) and aromatic (phenylpropanoids and polyphenols) compounds [32]. The main components of goji wax are alkanes, fatty acids, and alcohols [33]. GS-MS analysis showed that the wax components of ‘Ningqi I’ goji leaves and ‘Huangguo’ goji leaves included alkanes, alcohols, fatty acids, and esters (Figure 3B,C). Our further studies showed that the main components of wax in the leaves of the two species were basically the same, but the proportion of each wax component was different. The wax components with the highest proportion in the leaves of ‘Ningqi I’ goji and ‘Huangguo’ goji were alkanes, accounting for 82.6% and 65.9% of the total wax content (Figure 3B,C), respectively. This indicates that alkanes are the main components of the leaf wax layer. Similar results have also been reported for the drought shrub *Ammopiptanthus mongolicus* [31]. Linear alkanes have strong hydrophobicity [34], so we speculate that wax mainly composed of alkanes reduces the epidermal permeability of goji leaves. Previous studies have shown that the content of alkanes in the epidermal wax of rice (*Oryza sativa*), *Arabidopsis thaliana*, tobacco (*Nicotiana tabacum*), and *Ammopiptanthus mongolicus* is significantly high, which reduced the epidermal permeability and improved the drought resistance of plants [11,31]. Therefore, we believe that ‘Ningqi I’ goji has a higher drought resistance potential than ‘Huangguo’ goji.

Fatty acids and their derivatives are a class of compounds with complex functions. Fatty acids can participate in the wax synthesis pathway for the leaf epidermis of plants [35,36]. In this study, 11 differential metabolites, such as lipids (of which the main components are fatty acids), lactones, and flavonoids, were screened using UPLC-MS/MS analysis (Figure 4B and Appendix A). The analysis of the correlation between these differential metabolites and wax components showed that differential metabolites, such as phosphatidylcholine (PE), 6-Hydroxykaempferol, and DG (i-20:0/0:0/LTE4), were involved in the synthesis of wax components such as alkanes, alcohols, and fatty acids in the epidermis of goji leaves, which in turn made the total amount of wax in the epidermis of ‘Ningqi I’ goji leaves much higher than that of ‘Huangguo’ goji leaves (Figure 5). Studies have shown that *CYP704B1* regulates the conversion of C16 palmitic acid to 16-hydroxyhexadecanoic acid and promotes wax synthesis in the epidermis of barley (*Hordeum vulgare* L.) leaves [37]. This is similar to the results of our study.

The synthesis of plant cuticular wax is regulated by multiple genes [11,12]. We screened 23 genes related to wax synthesis through transcriptome data analysis and RT-qPCR, such as *LbaCER1*, *LbaKCS1*, and *LbaKCS11* (Figure 6 and Figure 8A,B). After combining these results with those of GC-MS data analysis, we found that the differential components of wax in the epidermis of ‘Ningqi I’ and ‘Huangguo’ goji leaves were mainly alkanes, fatty acids, and alcohols (Figure 3D–G). Alkanes are mainly synthesized by ultra-long-chain aldehyde decarbonylase (CER) [38,39]. In the alkane synthesis pathway, we identified three DEGs, specifically the down-regulated expression levels of two *LbaCER1* genes (Lba06g02387 and Lba01g00793) associated with ultra-long chain aldehyde decarbonylase 1. In contrast, the expression of the *LbaCER3* gene, associated with ultra-long chain aldehyde decarbonylase 3, was up-regulated (Figure 6). Previous studies have identified the functions of these genes in a variety of plants. For example, *BdCER1-8* clearly plays a dominant role in alkane biosynthesis in the epidermis of *Brachypodium distachyon* leaves [15]; *AtCER3* controls epidermal alkane synthesis in *Arabidopsis thaliana* [38]. RT-qPCR results show that the expression levels of *LbaCER1* (Lba06g02387 and Lba01g00793) in ‘Ningqi I’ goji leaves were significantly higher than those in ‘Huangguo’ goji leaves. However, the overall trend of RT-qPCR results was consistent with the transcriptome data, which may have been due to a false positive result from a single gene during the transcriptome experiment (Figure 8A,B). When these results were combined with GC-MS data, we found that the content of alkanes in the cuticular wax of ‘Ningqi I’ goji leaves was significantly higher than that of ‘Huangguo’ goji leaves (Figure 3B–E). Therefore, we speculate that, on the one hand, *LbaCER1* and *LbaCER3* are involved in the synthesis of alkanes; on the other hand, other enzymes and genes are involved in the synthesis of alkanes. In the previous study, the research group found a gene (Lba09g01421) encoding aldehyde decarbonylase (AD) in the alkane synthesis pathway, and its expression level in ‘Ningqi I’ goji leaves was significantly higher than in ‘Huangguo’ goji leaves [25]. This result supports our hypothesis that there is a complex gene network in ‘Ningqi I’ goji that regulates the synthesis of alkanes.

Transcriptomic analysis showed that the expression levels of *LbaKCS1*, *LbaKCS11*, and *LbaKCS20* in the leaves of ‘Ningqi I’ goji were significantly higher than those of ‘Huangguo’ goji (Figure 6). The content of C19 fatty acid in the leaf wax of ‘Ningqi I’ goji was also significantly higher than that of ‘Huangguo’ goji (Figure 3G). Previous studies have shown that KCS is a key enzyme in the wax synthesis pathway, which can catalyze the production of very long-chain fatty acids and promote wax accumulation [12,40]. Similarly, *LbaFAR2* (Lba11g02102) was also highly expressed in the leaves of ‘Ningqi I’ goji (Figure 6), and the content of C16 alcohol in the leaf wax of ‘Ningqi I’ goji was significantly higher than that of ‘Huangguo’ goji (Figure 3F). Previous studies have also shown that FAR is a key gene in the acyl reduction pathway, which promotes alcohol biosynthesis [11,41]. Therefore, we speculate that *LbaKCS1*, *LbaKCS11*, *LbaFAR2* (Lba11g02102), and *LbaKCS20* may be important genes involved in wax synthesis.

MYB and WRKY transcription factors can regulate plant epidermal wax synthesis. For example, wax accumulation caused by the overexpression of *OsWRKY89* in the epidermis of rice plant leaves increased, while wax accumulation, due to RNAi-mediated *OsWRKY89*-silencing lines, decreased in the epidermis of rice plant leaves [42]. In this study, we found that the expression levels of six transcription factors, namely *LbaMYBS3*, *LbaMYB48*, *LbaMYB3*, *LbaMYB306*, *LbaMYB60*, and *LbaWRKY68*, were significantly up-regulated in ‘Ningqi I’ goji leaves (Figure 7 and Figure 8C–H), which corresponded with the trend of high wax content in the epidermis of ‘Ningqi I’ goji leaves (Figure 11). Therefore, we speculated that *LbaMYBS3*, *LbaMYB48*, *LbaMYB3*, *LbaMYB306*, *LbaMYB60*, and *LbaWRKY68* may be involved in the wax synthesis of goji leaves.

Although this study provides an in-depth insight into the wax synthesis mechanism of goji leaves, there are still some limitations. For example, we have not yet fully evaluated all the genes involved in wax synthesis, and the functional verification of these genes has not yet been completed. Future research should focus on the functional verification of these key genes and drought resistance experiments of transgenic lines, with a view to providing new strategies and research directions for the genetic improvement and stress resistance management of goji plants.

## 4. Materials and Methods

### 4.1. Plant Materials

The experimental plants were 10-year-old ‘Ningqi I’ goji (*Lycium barbarum* cv. ‘Ningqi I’) and ‘Huangguo’ goji (*Lycium barbarum* var. auranticarpum K. F. Ching) seedlings. All were planted at the Economic Forest Cultivation Base of the College of Forestry, Gansu Agricultural University (36°09′11″ N, 103°70′15″ E, altitude 1581.4 m). The seedlings grew under the same management conditions, with organic fertilizer applied once a year in winter (10 kg per plant), pruned once in winter, and irrigated once a month from March to October. The average height of the ‘Ningqi I’ goji trees was 1.73 m, with an average ground diameter of 48.6 mm; the average height of the ‘Huangguo’ goji trees was 1.69 m, with an average ground diameter of 46.7 mm. In the experiment, all leaves were collected from one-year-old branches at the same growth node, namely, fresh, mature leaves that were free from diseases and pests.

### 4.2. Analysis and Comparison of Leaf Phenotype, Water Loss Rate, and Chlorophyll Leaching Rate

The determination of the water loss rate was carried out by referring to the method of T. N. McCaig [43], with some modifications, as follows: the leaves of ‘Ningqi I’ goji and ‘Huangguo’ goji were shaded for 6 h to close the stoma. Then, the leaves were collected and placed in the dark (at 25 °C and a relative humidity of 75%) for dehydration. Each cultivar was weighed using an electronic balance at 3, 6, 12, 24, and 36 h after separation, with 15 leaves used as biological replicates. The water loss rate was expressed as a percentage of the weight lost in comparison with the initial leaf weight.

The determination of the chlorophyll leaching rate was conducted according to a previous method [44]. Thirty fully expanded leaves were taken from ‘Ningqi I’ goji and ‘Huangguo’ goji, respectively, and three biological replicates were set. These leaves were soaked in 80% ethanol and stored in a dark environment (at 25 °C with a relative humidity of 75%). After initial soaking for 2, 6, 12, 24, 36, and 48 h, the absorbance of the extract at 647 nm (A647) and 664 nm (A664) wavelengths was determined using a spectrophotometer (UV-2600, Shimadzu, Kyoto, Japan), with 1 mL of solution used. The calculation formula for chlorophyll concentration was as follows: total micromolar chlorophyll = 7.93 × A664 + 19.53 × A647 (Ritchie, R. J.; [45]). The chlorophyll leaching rate was expressed as the percentage of chlorophyll concentration at each time point in the total chlorophyll extracted after 48 h.

### 4.3. Scanning Electron Microscopy (SEM) Analysis

For this step, we referred to Pathan et al.’s method [46]. Leaves with the same node, healthy growth, and no pests and diseases were collected from the goji plants as sampling materials. The samples were cut into flakes measuring approximately 1 cm^2^ and treated briefly with frozen methanol in liquid nitrogen. Before observing the wax crystals on the leaf epidermis, the frozen samples were freeze-dried for 24 h. Each leaf sample was extensively scanned under a scanning electron microscope at two different magnifications (50× and 100×).

### 4.4. Analysis of Wax Content and Composition in Leaves

The mature leaves of ‘Ningqi I’ goji and ‘Huangguo’ goji were used as sampling materials. Firstly, the leaf area was scanned using a scanner and calculated using Image J 1.53a software. Then, the leaves were immersed in 15~20 mL of chloroform solution, which was gently shaken to ensure the leaves were fully immersed. After 45 s, the leaves were taken out of the solution, and 15 μL of C24 alkanes (1 g/mL) was added to the extract as an internal reference. The solution was then filtered using a funnel. When it naturally volatilized to 0.5~1.0 mL, the wax was transferred to the gas chromatography sample bottle and dried using nitrogen. To dissolve the wax, 35 μL of pyridine was added to each sample bottle; then, an equal volume of BSTFA [N, O-bis (trimethylsilyl) trifluoroacetamide] was added. The bottle cap was quickly covered and the bottles were placed in a water bath at 70 °C for 1 h. Then, the derivative reagent was dried using a nitrogen constant temperature drier at 70 °C, and 1.5 mL of chloroform was added to dissolve it. The pore size was 0.22 μm. The syringe microporous membrane was then filtered into a 1.5 mL headspace injection bottle for GC-MS analysis.

For the determination of the wax components, we referred to a previous method [47]. Wax component determination was carried out using Thermo DSQ II gas chromatography–mass spectrometry (Thermo Fisher Scientific, Waltham, MA, USA). A DB-5 capillary column (30 m × 0.25 mm × 0.25 μm) was used for component separation, helium (He) was used as the carrier gas, and the flow rate was 1 mL/min. The gas phase program was set as follows: the initial temperature was maintained at 70 °C for 1 min; the temperature was increased to 200 °C at the speed of 10 °C/min; then, the temperature was increased to 300 °C at 4 °C/min and maintained for 20 min. The split ratio was 3/1, and the injection volume was 1 μL. The mass spectrometry program settings was as follows: the inlet temperature was 250 °C; the transmission line temperature was 250 °C; the ion source temperature was 230 °C; the electron ionization (EI) source was 70 eV; and the mass spectrometer’s scanning range was 50–650 *m*/*z*. Wax components were identified according to the National Institute of Standards and Technology (NIST) 09 mass spectrometry library. The wax compounds were quantified by comparing the peak area of the wax compounds with the internal standard, and the content was expressed in the form of μg/cm^2^. Each variety has three biological components.

### 4.5. Ultra-Performance Liquid Chromatography–Tandem Mass Spectrometry (UPLC-MS/MS) Analysis of Leaves

UPLC-MS/MS analysis was carried out according to a previous method [48]. Samples weighing 60 mg were placed into a 1.5 mL centrifuge tube, and two small steel balls and 600 μL of a methanol–water mixture (V/V = 7/3, containing mixed internal standard, 4 μg/mL) were added. After being pre-cooled in a refrigerator at −40 °C for 2 min, they were ground in a grinder (at 60 Hz for 2 min). Following an ultrasonic extraction in an ice water bath for 30 min, they were placed at −40 °C overnight. After undergoing centrifugation at 4 °C for 10 min (at 12,000 rpm), 150 μL of supernatant was extracted with a syringe, filtered using a 0.22 μm organic phase pinhole filter, transferred to the LC injection vial, and stored at −80 °C until LC-MS analysis. The analytical instrument used in this experiment was a liquid chromatography–mass spectrometry system composed of Waters ACQUITY UPLC I-Class plus ultra-high performance liquid chromatographer (Waters Corporation, Waltham, MA, USA) and Thermo QE tandem high resolution mass spectrometer (Thermo Fisher Scientific, Waltham, MA, USA). The operating parameters were as follows: chromatographic column: ACQUITY UPLC HSS T3 (100 mm × 2.1 mm, 1.8 μm); column temperature: 45 °C; mobile phase: A-water (containing 0.1% formic acid) and B-acetonitrile; flow rate: 0.35 mL/min; and injection volume: 3 μL. This experiment was set up with six biological replicates.

### 4.6. Real-Time Quantitative PCR (RT-qPCR)

Through comparative analysis, six genes related to wax synthesis were screened out. The leaves of ‘Ningqi I’ goji and ‘Huangguo’ goji were analyzed using RT-qPCR. The total RNA was extracted using a nucleic acid purification kit (Beijing Tiangen Biotechnology Co., Ltd., Beijing, China) and reverse transcribed using a HiScrip III first-strand cDNA synthesis kit (+gDNA removal) (Vazyme, Nanjing, China). The primers were designed using Primer 5.0 and Oligo 7.0 software in the conserved region of the gene (Shanghai Bioengineering Technology Service, Shanghai, China). The primers are shown in the Appendix A (Appendix A). RT-qPCR amplification was performed with 20 μL of reaction solution, containing 1 μL of upstream and downstream primers, 6 μL of water, 2 μL of cDNA, and 10 μL of 2 × CHAMQ SYBR COLOR QPCRMaster Mix (Vazyme, Nanjing, China). RT-qPCR was detected using the Light Cycler^®^480 instrument (Basel Roche, Basel, Switzerland). Transcription levels were measured using the 2^−ΔΔCT^ method [49]. Each gene was repeated three times.

### 4.7. Statistical Analysis

All data are shown as the means ± standard deviations. Student’s *t*-test, in IBM SPSS Statistics 26 software, was used to compare the data differences between ‘Ningqi I’ goji and ‘Huangguo’ goji. The results were considered statistically significant at the *p* < 0.01 and the *p* < 0.05 levels.

## 5. Conclusions

The total wax content and wax components (alcohols, fatty acids, and alkanes) in ‘Ningqi I’ goji leaves are significantly higher than those in ‘Huangguo’ goji leaves. This increase in wax content has reduced the permeability of the cuticle in the leaves of ‘Ningqi I’ goji, thereby reducing the water loss rate of the goji leaves, and ultimately enhancing the drought resistance potential of ‘Ningqi I’ goji plants. Transcriptomic and metabolomic analyses revealed that the presence of wax plays a crucial role in mediating drought resistance in goji plants. The up-regulation of key genes (such as *LbaWSD1* and *LbaCER1*) and transcription factors (such as *LbaMYB3*) involved in wax synthesis can facilitate the accumulation of waxy components, such as alcohols (C16), fatty acids (C19), and alkanes (C31 and C33), to enhance the plant’s stress tolerance.

## Figures and Tables

**Figure 1 ijms-25-10939-f001:**
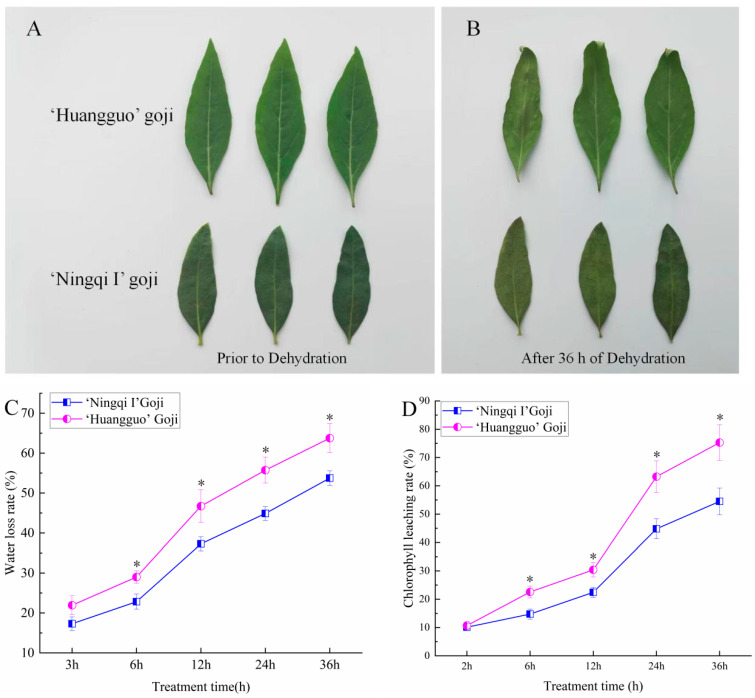
Analysis of leaf phenotype and cuticle permeability of ‘Ningqi I’ goji and ‘Huangguo’ goji. (**A**) Photos of ‘Ningqi I’ goji leaves and ‘Huangguo’ goji leaves before dehydration; (**B**) photos of ‘Ningqi I’ goji leaves and ‘Huangguo’ goji leaves after dehydration; (**C**) comparison of dehydration rate of ‘Ningqi I’ goji and ‘Huangguo’ goji leaves after dehydration for 36 h; (**D**) comparison of chlorophyll leaching rate between ‘Ningqi I’ goji and ‘Huangguo’ goji leaves after alcohol soaking for 36 h. Vertical bars represent standard deviations of the means (*n* = 3). Significant differences between ‘Ningqi I’ goji and ‘Huangguo’ goji leaves at the *p* < 0.05 levels were indicated by ‘*’, according to Student’s *t*-test.

**Figure 2 ijms-25-10939-f002:**
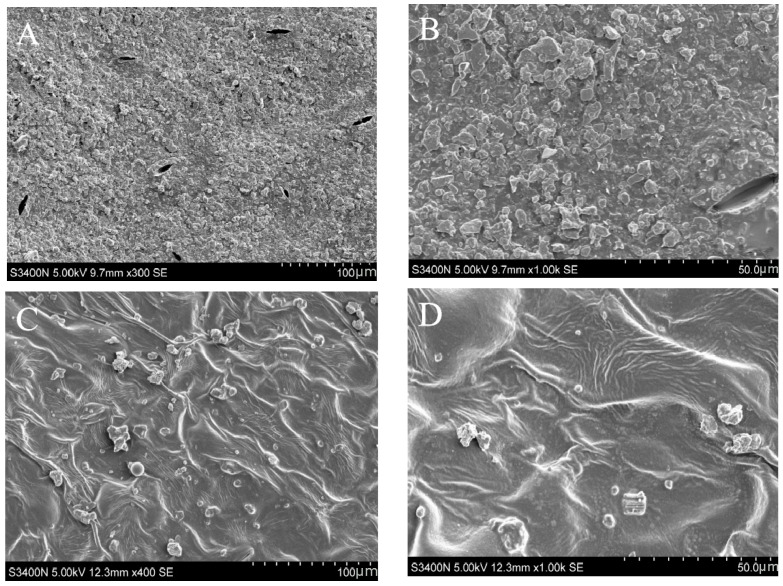
Scanning electron micrograph of wax on surfaces of ‘Ningqi I’ goji and ‘Huangguo’ goji leaves. (**A**) A 400× magnification of ‘Ningqi I’ goji leaf epidermis wax, (**B**) a 1000× magnification of ‘Ningqi I’ goji leaf epidermis wax, (**C**) a 400× magnification of ‘Huangguo’ goji leaf epidermis wax, and (**D**) a 1000× magnification of ‘Huangguo’ goji leaf epidermis wax.

**Figure 3 ijms-25-10939-f003:**
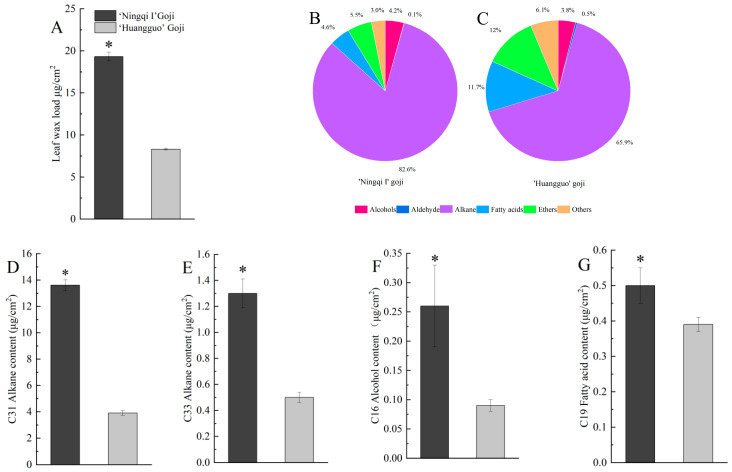
Total wax content and wax components of ‘Ningqi I’ goji leaves and ‘Huangguo’ goji leaves. (**A**) Total wax content of ‘Ningqi I’ goji leaves and ‘Huangguo’ goji leaves; (**B**) wax components of ‘Ningqi I’ goji leaves; (**C**) wax components of ‘Huangguo’ goji leaves; (**D**) tricosane content; (**E**) tricosane content; (**F**) phytol content; (**G**) nonadecanoic acid content. ‘*’ represents extremely significant differences (*p* ≤ 0.05).

**Figure 4 ijms-25-10939-f004:**
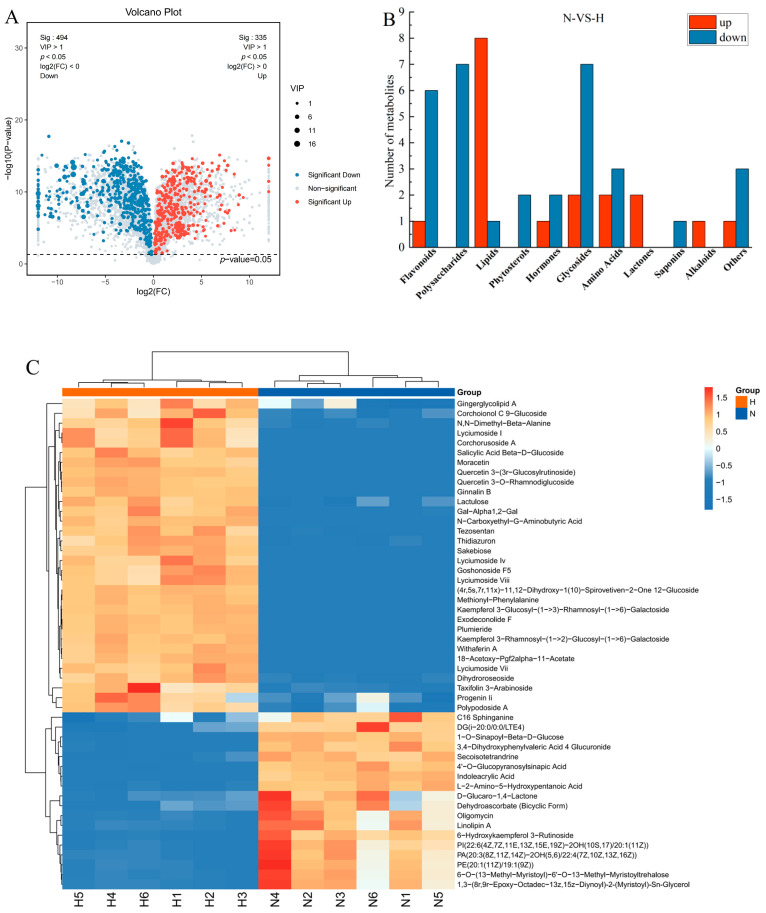
Analysis of differential metabolites in leaves of ‘Ningqi I’ goji and ‘Huangguo’ goji. (**A**) Top 50 differential metabolites shown using volcano plot; (**B**) differential metabolites classification plot; and (**C**) cluster heat map. ‘N’ represents ‘Ningqi I’ goji and ‘H’ represents ‘Huangguo’ goji.

**Figure 5 ijms-25-10939-f005:**
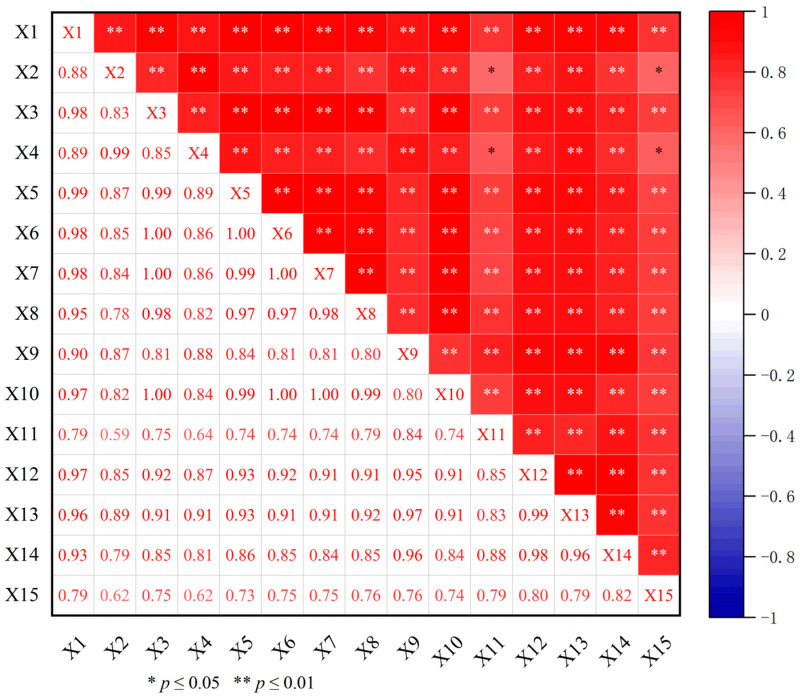
Correlation analysis between dominant components of wax and differential metabolites. X1–X15 represent 6-Hydroxykaempferol 3-Rutinoside, D-Glucaro-1,4-Lactone, 6-O-(13-Methyl-Myristoyl)-6′-O-13-Methyl-Myristoyltrehalose, Dehydroascorbate (Bicyclic Form), PI(22:6(4Z,7Z,11E,13Z,15E,19Z)-2OH(10S,17)/20:1(11Z)), PA(20:3(8Z,11Z,14Z)-2OH(5,6)/22:4(7Z,10Z,13Z,16Z)), PE(20:1(11Z)/19:1(9Z)), Linolipin A, DG(i-20:0/0:0/LTE4), 1,3-(8r,9r-Epoxy-Octadec-13z,15z-Dien-4,6-Diynoyl)-2-(Myristoyl)-Sn-Glycerol, C16 Sphinganine, C31 alkanes, C33 alkanes, C16 alcohols, and C19 fatty acids. ‘N’ stands for ‘Ningqi I’ goji leaves and ‘H’ represents ‘Huangguo’ goji leaves. Note: ‘*’ represents a significant positive correlation (*p* ≤ 0.05); ‘**’ represents a very significant positive correlation (*p* ≤ 0.01).

**Figure 6 ijms-25-10939-f006:**
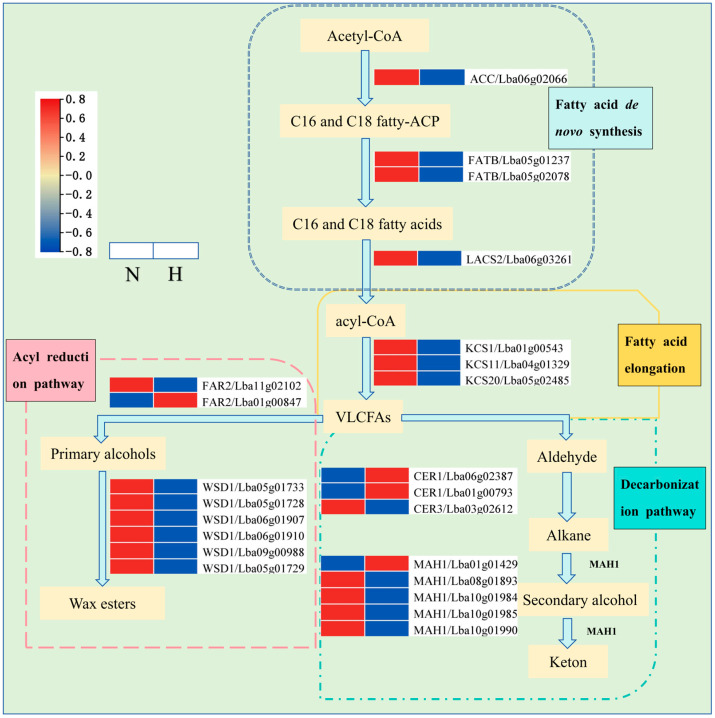
Transcription and expression analysis of genes related to wax synthesis pathway in leaves of ‘Ningqi I’ goji and ‘Huangguo’ goji. Main pathways of plant wax biosynthesis are fatty acid de novo synthesis, fatty acid elongation, decarbonylation pathway, and acyl reduction pathway. Expression level gradually increased from blue to red. Very long chain fatty acids (VLCFAs); acetyl-CoA carboxylase (ACC); palmitoyl carrier protein thioesterase (FATB); long-chain acyl-CoA synthase 2 (LACS2); β-ketoacyl-CoA synthase (KCS); ultralong-chain aldehyde decarbonylase 1 (CER1); alkane hydroxylase (MAH1); ultralong-chain aldehyde decarbonylase 3 (CER3); fatty acyl-CoA reductase 2 (FAR2); wax ester synthase (WSD1). ‘N’ stands for ‘Ningqi I’ goji leaves and ‘H’ represents ‘Huangguo’ goji leaves.

**Figure 7 ijms-25-10939-f007:**
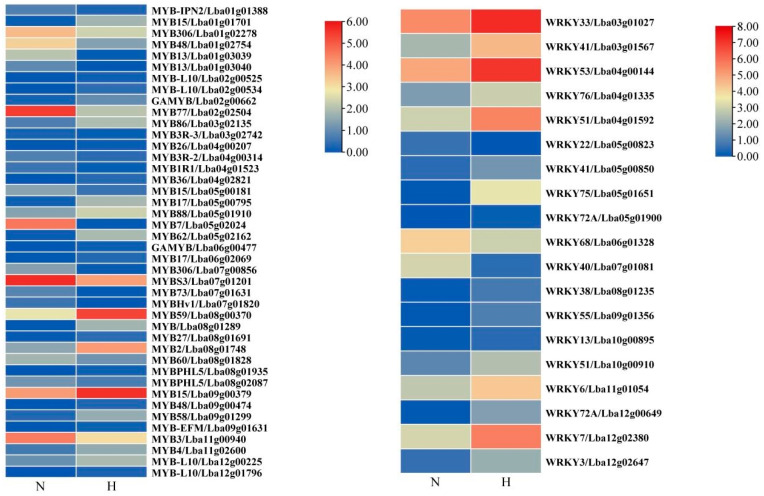
MYB transcription factor FPKM value heat map and WRKY transcription factor FPKM value heat map. ‘N’ stands for ‘Ningqi I’ goji leaves and ‘H’ represents ‘Huangguo’ goji leaves.

**Figure 8 ijms-25-10939-f008:**
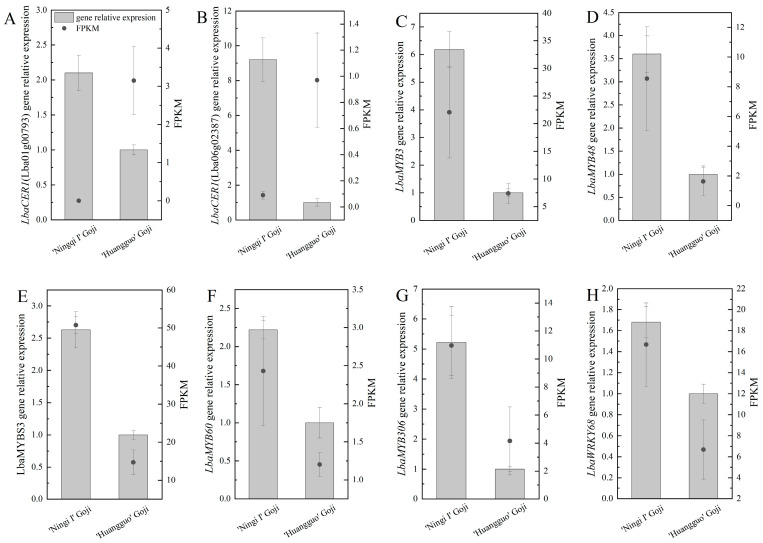
Real-time quantitative PCR analysis of DEGs and TFs related to wax synthesis. Relative expression levels of (**A**) *LbaCER1* (Lba06g00793), (**B**) *LbaCER1* (Lba06g02387), (**C**) *LbaMYB3*, (**D**) *LbaMYBS3*, (**E**) *LbaMYB48*, (**F**) *LbaMYB60*, (**G**) *LbaMYB306*, and (**H**) *LbaWRKY68*. Expression levels in ‘Huangguo’ goji leaves were used as a reference.

**Figure 9 ijms-25-10939-f009:**
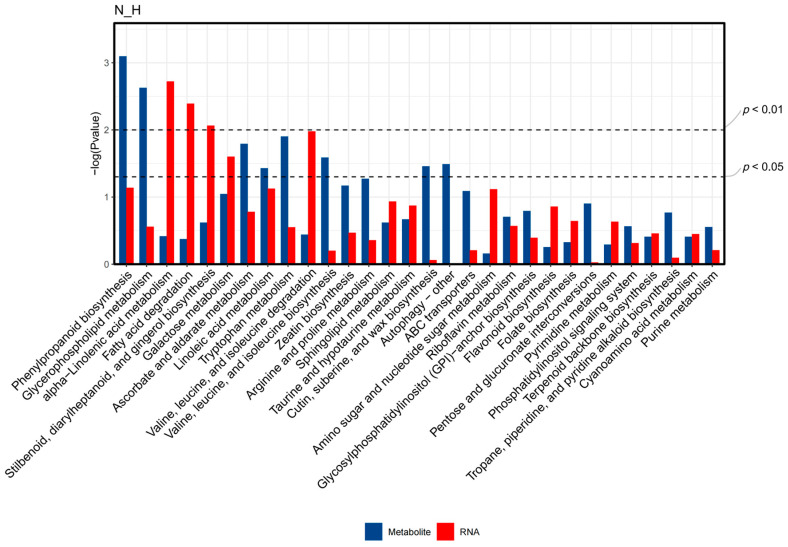
Pathway histogram of the common mappings of 30 pathway differentials with the smallest sum of *p* value. The abscissa is the pathway, and the ordinate is the *p*-value of the pathway enrichment. (*p* ≤ 0.05) represents significant enrichment and (*p* ≤ 0.01) represents extremely significant enrichment.

**Figure 10 ijms-25-10939-f010:**
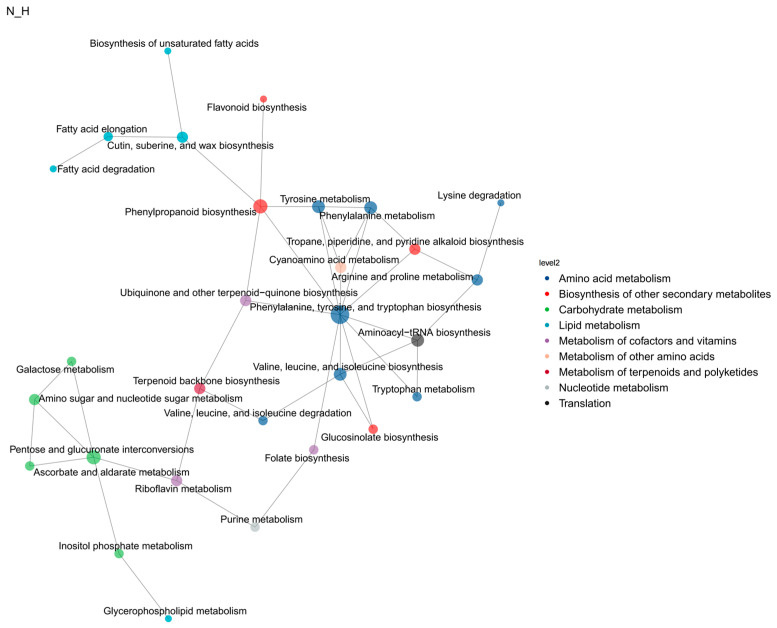
KGML interaction network diagram.

**Figure 11 ijms-25-10939-f011:**
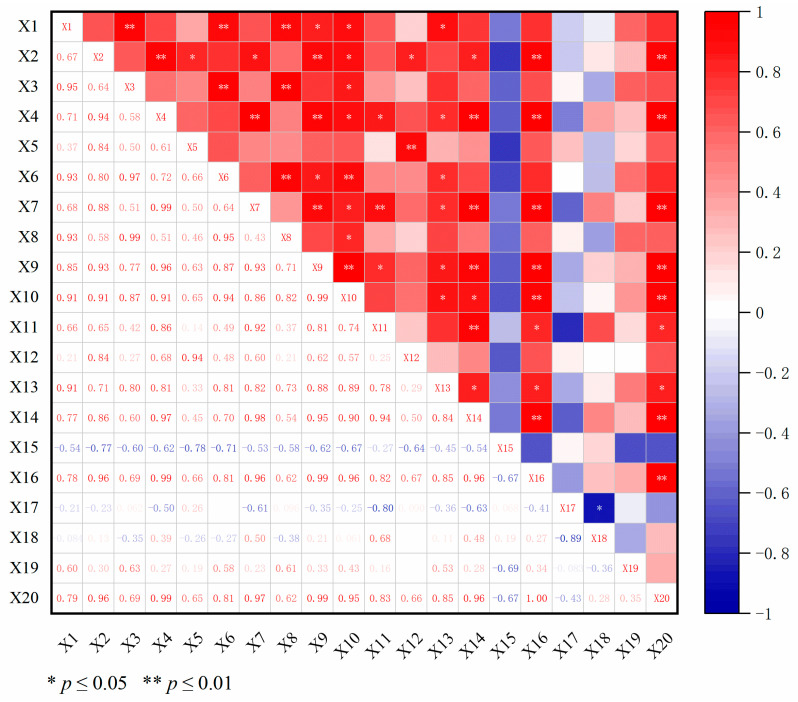
Correlation heat map of wax synthesis-related genes and transcription factors with total wax content and components. X1–X20 represent *LbaFATB*/Lba05g02078, *LbaLACS2*/Lba06g03261, *LbaFAR2*/Lba11g02102, *LbaWSD1*/Lba05g01733, *LbaWSD1*/Lba05g01728, *LbaWSD1*/Lba09g00988, *LbaWSD1*/Lba05g01729, *LbaMYB3*/Lba11g00940, *LbaMYBS3*/Lba07g01201, *LbaMYB306*/Lba07g00856, *LbaMYB60*/Lba08g01828, *LbaMYB48*/Lba01g02754, *LbaWRKY68*/Lba06g01328, alcohols, aldehydes, alkanes, fatty acids, esters, others and total wax content. ‘*’ represents a significant positive correlation (*p* ≤ 0.05) and ‘**’ represents a very significant positive correlation (*p* ≤ 0.01).

## Data Availability

To support the results, the transcriptome data set has been deposited in the GSA database (https://ngdc.cncb.ac.cn/gsa, accessed on 11 March 2024), with alogin accession of CRA015284.

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
