# Peer review of "Integration of Transcriptome and Metabolome Reveals Wax Serves a Key Role in Preventing Leaf Water Loss in Goji (Lycium barbarum)"

_ijms, 2024, doi:10.3390/ijms252010939_

Round 1

Reviewer 1 Report

Comments and Suggestions for Authors

To: MDPI IJMS Editorial Office

Please find my review for the manuscript ID: ijms-3162504 entitled as “Wang et al., Integration of Transcriptome and Metabolome Reveals wax Serves as a key role in Preventing leaf Water loss in Goji (Lycium barbarum)” with the objectives: to investigate the relationship of water loss and wax formation mechanism in Goji (Lycium barbarum) leaves targeting two contrasting genotypes of goji namely ‘Ningqi I’ goji and ‘Huangguo’ goji as study materials. The authors have analyzed the differences in water loss rate, chlorophyll leaching rate, wax phenotype, wax content and components in two tested materials. The differences in wax synthesis related pathways transcriptome and metabolome were analyzed in relation to wax components, wax synthesis genes and transcription factors. Based on the study, the manuscript is well written. The introduction part clearly reviewed with coherent information. Pertinent data with figures were generated. The results and discussion parts of the text were well articulated with sound interpretation.   

However, I would like to mention the following limitations or issues with regarding this manuscript that should be amended.   

1.     I have seen that the figures generated in this study were not fully utilized or cited for the understanding of the reader. In addition, there were labels missing of some. There is a problem of coordination of the generated figures with the text in the results and discussion parts of the manuscript. For example:

a.      Figure 2 A, B, C & D labels missing and not well cited/referred in the text (results and discussion) parts. Therefore, it is very difficult to identify which figures representing which part of the article and also very hard to synchronize the text with the figures.

b.     Figure 3 B & C labels missing. In addition, the whole figure 3 A-G were never ever referred or cited in the article (result & discussion parts) at all. So, what is the important of these figures and why they have been presented here?

c.      Figure 6 also never ever cited at all too! It is better to delete from the article.

d.     The holds true for figure 8 B-H. They only mentioned figure 8A while neglecting figure 8 B-H.

e.      Figure 4 A & C labels missing.

f.      The same holds true for supplementary figure S3 in missing of the labels and citation in the text.

g.      Please check all figures coordination throughout the manuscript.

2.     In summary, generated figures were not well cited and elaborated in the text. I don’t know why the authors didn’t give due attention for the values of the figures they have generated. Surprisingly, out of 11 main figures and three supplemental figures presented in this manuscript, a single figure was not cited in the discussion part of the article!

3.     Correlation coefficient values were not cited or missing in the text line#424-435.

4.     Is it “2-3 types” or “23 types” at line #465?

5.     Check whether “LbaMYBS3” or “LbaMYB53” line#29, 371, 386, 401, 428, 551 and 555. Mostly MYB TF followed by different given number. Or are you talking other TF named “MYBS3”?

Thank you!

Author Response

Dear Editors and Reviewers,

Thank you for your kind work and for the reviewers’ comments concerning our manuscript entitled “Integration of Transcriptome and Metabolome Reveals wax Serves as a key role in Preventing leaf Water loss in Goji (Lycium barbarum)”. Those comments are all valuable and very helpful for revising and improving our paper, as well as the important guiding significance to our researches. We have studied comments carefully and have made correction. Revised portion are marked in red in the paper. The main corrections in the paper and the responses to reviewers’ comments are as follows:

Reviewer #1:

  1. I have seen that the figures generated in this study were not fully utilized or cited for the understanding of the reader. In addition, there were labels missing of some. There is a problem of coordination of the generated figures with the text in the results and discussion parts of the manuscript. For example:
  2. Figure 2 A, B, C & D labels missing and not well cited/referred in the text (results and discussion) parts. Therefore, it is very difficult to identify which figures representing which part of the article and also very hard to synchronize the text with the figures.

Response: Thank you very much for reviewers’ valuable advice, Figure 2 A, B, C, and D have had complete labels added, and they have been cited/referred to in the text (Results and Discussion). The cited parts have been highlighted in red.

  1. Figure 3 B & C labels missing. In addition, the whole figure 3 A-G were never ever referred or cited in the article (result & discussion parts) at all. So, what is the important of these figures and why they have been presented here?

Response: According to the reviewer's opinion, I have added complete labels to Figure 3 and highlighted the references in the text with red.

  1. Figure 6 also never ever cited at all too! It is better to delete from the article.

Response: Thank you very much for reviewers’ valuable advice, Figure 6 is mainly used to illustrate the key genes related to wax synthesis that were identified through transcriptome screening.Figure 6 has been cited in the article and marked with red.

  1. The holds true for figure 8 B-H. They only mentioned figure 8A while neglecting figure 8 B-H.

Response: Thank you very much for reviewers’ valuable advice, I have cited the other parts of Figure 8 and highlighted them in red within the article.

  1. Figure 4 A & C labels missing.

Response: Thank you very much for reviewers’ valuable advice, I have added complete labels to Figure 4.

  1. The same holds true for supplementary figure S3 in missing of the labels and citation in the text.

Response: Thank you very much for reviewers’ valuable advice, I have added the full tags for supplementary Figure S3 and highlighted them in red when referenced in the article.

  1. Please check all figures coordination throughout the manuscript.

Response: Thank you very much for reviewers’ valuable advice, I have carefully checked the coordination of all figures throughout the manuscript, ensuring consistency in format and content. Additionally, I have reviewed the labels and titles of the figures to ensure they are clear, accurate, and closely related to the text content.

  1. In summary, generated figures were not well cited and elaborated in the text. I don’t know why the authors didn’t give due attention for the values of the figures they have generated.Surprisingly, out of 11 main figures and three supplemental figures presented in this manuscript, a single figure was not cited in the discussion part of the article!

Response: Thank you very much for reviewers’ valuable advice, I have added labels of all the charts and graphs in the paper as required and quoted them in the conclusion and discussion of the paper. The numbers of the quoted charts and graphs have been marked in red in the paper.

  1. Correlation coefficient values were not cited or missing in the text line#424-435.

Response: Thank you very much for reviewers’ valuable advice, I have quoted the value of the correlation coefficient in the article. It has now been changed to 450-459 lines.

  1. Is it “2-3 types” or “23 types” at line #465?

Response: Thank you very much for reviewers’ valuable advice, It is “23 types” at line #465.For more accurate expression, I have rewritten the corresponding sentences and highlighted them in red. It has now been changed to 499-500 lines.

  1. Check whether “LbaMYBS3” or “LbaMYB53” line#29, 371, 386, 401, 428, 551 and 555. Mostly MYB TF followed by different given number. Or are you talking other TF named “MYBS3”?

Response: Thank you very much for reviewers’ valuable advice,The name of the transcription factor in lines 29, 371, 386, 401, 428, 551 and 555 is "LbaMYBS3". I have checked and made sure its name is correct.

Yours Sincerely,

Xingbin Wang

Jie Li

Reviewer 2 Report

Comments and Suggestions for Authors

Dear Editors and Authors, I read with interest the manuscript entitled Integration of Transcriptome and Metabolome Reveals wax Serves as a key role in Preventing leaf Water loss in Goji (Lycium barbarum). In this study, ‘Ningqi I’ goji and ‘Huangguo’ goji, with significant differences in leaf wax content were used as materials to determine the differences in leaf water loss rate and chlorophyll leaching rate between the two types of goji. The differences in the morphology and chemical composition of leaf wax were compared, the wax metabolism pathway and biosynthesis-related genes in leaves were analyzed, and a correlation analysis was carried out. The subject of the article is important and has great relevance for the scientific environment of the study area. Therefore, the manuscript needs some adjustments so that it can then be forwarded to the publication process. The manuscript has the potential for publication in this journal International Journal of Molecular Sciences and needs the following adjustments:

ABSTRACT

- Separate the word “Drought” from the “:”.

- Mention the process of transpiration when mentioning water loss through leaves. This was not done.

- Are 'Ningqi I' goji and 'Huangguo' goji cultivars? Mention this.

- Keywords: Replace repeated terms in the title. This will help the article to be found in a future publication.

INTRODUCTION

- At the beginning of the Second paragraph, mention that it is “cuticular wax”.

- Mention the name of the classifier, along with the scientific name of the species.

- Include some hypotheses before mentioning the objective. This will enrich the article.

- Mention the objective as follows: the aim of the study ... It was not cited in this way, and this may make it difficult for the reader to understand.

- In the last paragraph, what was analyzed is mentioned. I suggest removing it. This is Material and Methods.

MATERIAL AND METHODS

- Insert geographic coordinates of the study site.

- What were the water, fertilizer and pruning management methods used? This needs to be detailed in the first paragraph.

- What is the size (height and diameter) of these plants after 14 years of cultivation? Give more details.

- The water loss rate was calculated according to which methodology? Cite reference.

- The leaves collected were from which part of the plant? Middle third? What were the sanitary conditions of the plants at the time? Collection? Healthy, damage-free plants?

- Who proposed the formula for calculating total micromolar chlorophyll? Cite.

- Split paragraph of topic 2.4.

- The last section of topic 2.7. needs to be rewritten. See examples in the articles. You need to inform that the analysis was at 5% and 1% probability.

RESULTS

- In topic 3.1. it was stated that the epidermal permeability of 'Huangguo' goji leaves was significantly higher than that of 'Ningqi I' goji leaves. How was this measured? Was it possible to make this claim based solely on the images of the leaves?

- Figure 2 should be presented after the first citation in the text. Move.

- The rest of this section is well-written, and the figures are of good quality.

DISCUSSION

- There is repeated information in the Introduction.

- The first excerpt has already been cited in the Introduction, where it is stated that drought stress is detrimental to the growth and quality of the fruits.

- Some information is not associated with the results found.

- Several excerpts are only about the theory (literature review) of a particular subject of the study. This needs to be revised.

CONCLUSION

- Do not start with “In summary”.

Author Response

Dear Editors and Reviewers,

Thank you for your kind work and for the reviewers’ comments concerning our manuscript entitled “Integration of Transcriptome and Metabolome Reveals wax Serves as a key role in Preventing leaf Water loss in Goji (Lycium barbarum)”. Those comments are all valuable and very helpful for revising and improving our paper, as well as the important guiding significance to our researches. We have studied comments carefully and have made correction. Revised portion are marked in red in the paper. The main corrections in the paper and the responses to reviewers’ comments are as follows:

Reviewer #2:

ABSTRACT

  1. Separate the word “Drought” from the “:”

Response: Thank you very much for reviewers’ valuable advice, I have added a space between "Drought" and ":".

  1. Mention the process of transpiration when mentioning water loss through leaves. This was not done.

Response: Thank you very much for reviewers’ valuable advice, I have written and marked in red the process of transpiration as requested.

3.Are 'Ningqi I' goji and 'Huangguo' goji cultivars? Mention this.

Response: Thank you very much for reviewers’ valuable advice, 'Ningqi I 'goji is a cultivated variety selected from Lycium barbarum L. . 'Huangguo' goji is a variety of wolfberry in Ningxia and is also used as a cultivar for production.I have mentioned this in the abstract of the article and highlighted it in red.

4.Keywords: Replace repeated terms in the title. This will help the article to be found in a future publication.

Response: Thank you very much for reviewers’ valuable advice, I have replaced the repeated terms in the title as requested and highlighted them in red.

INTRODUCTION

1.At the beginning of the Second paragraph, mention that it is “cuticular wax”.

Response: Thank you very much for reviewers’ valuable advice,I have changed "cuticular wax" to "wax".

2.Mention the name of the classifier, along with the scientific name of the species.

Response: Thank you very much for reviewers’ valuable advice, I have added the scientific names corresponding to the species in the introduction section and highlighted them in red.

  1. Include some hypotheses before mentioning the objective. This will enrich the article.

Response: Thank you very much for reviewers’ valuable advice, I have presented the hypothesis as requested and highlighted it in red before referring to the objective.

  1. Mention the objective as follows: the aim of the study ... It was not cited in this way, and this may make it difficult for the reader to understand.

Response: Thank you very much for reviewers’ valuable advice, I have rewritten the sentence as required and highlighted it in red in the introduction section.

  1. In the last paragraph, what was analyzed is mentioned. I suggest removing it. This is Material and Methods.

Response: Thank you very much for reviewers’ valuable advice, I have removed the analysis part from the introduction.

MATERIAL AND METHODS

  1. Insert geographic coordinates of the study site.

Response: Thank you very much for reviewers’ valuable advice, I have inserted the geographical coordinates of the study site.

  1. What were the water, fertilizer and pruning management methods used? This needs to be detailed in the first paragraph.

Response: Thank you very much for reviewers’ valuable advice, In the first paragraph, I added in detail the management methods of water, fertilizer and pruning.

  1. What is the size (height and diameter) of these plants after 14 years of cultivation? Give more details.

Response: Thank you very much for reviewers’ valuable advice, I added the average height and average diameter of the plant in the first paragraph.

4.The water loss rate was calculated according to which methodology? Cite reference.

Response: Thank you very much for reviewers’ valuable advice, I have cited the method of T. N. McCaig as a reference.

  1. The leaves collected were from which part of the plant? Middle third? What were the sanitary conditions of the plants at the time? Collection? Healthy, damage-free plants?

Response: Thank you very much for reviewers’ valuable advice, I have provided a detailed description of the sampling details in the first paragraph of the Materials and Methods section.

6.Who proposed the formula for calculating total micromolar chlorophyll? Cite.

Response: Thank you very much for reviewers’ valuable advice, I have cited Ritchie, R. J. 's method as a reference and highlighted it in the article.

7.Split paragraph of topic 2.4.

Response: Thank you very much for reviewers’ valuable advice, I have broken up the paragraph in section 2.4.

8.The last section of topic 2.7. needs to be rewritten. See examples in the articles. You need to inform that the analysis was at 5% and 1% probability.

Response: Thank you very much for reviewers’ valuable advice, I have made the required changes and highlighted them in red.

RESULTS

  1. In topic 3.1. it was stated that the epidermal permeability of 'Huangguo' goji leaves was significantly higher than that of 'Ningqi I' goji leaves. How was this measured? Was it possible to make this claim based solely on the images of the leaves?

Response: Thank you very much for reviewers’ valuable advice, The comprehensive results of leaf phenotype, water loss rate measurement, and chlorophyll leaching rate measurement experiments indicate that the epidermal permeability of "Huangguo" goji leaves is significantly higher than that of "Ningqi I" goji leaves. These specific results are detailed in Section 3.1, and I have highlighted them in red.

  1. Figure 2 should be presented after the first citation in the text. Move.

Response: Thank you very much for reviewers’ valuable advice, I have already placed Figure 2 in the appropriate position.

  1. The rest of this section is well-written, and the figures are of good quality.

Response: Thank you very much for your appreciation and recognition.

DISCUSSION

  1. There is repeated information in the Introduction.

Response: Thank you very much for reviewers’ valuable advice, The repeated part has been modified.

  1. The first excerpt has already been cited in the Introduction, where it is stated that drought stress is detrimental to the growth and quality of the fruits.

Response: Thank you very much for reviewers’ valuable advice, The repeated part has been modified.

  1. Some information is not associated with the results found.

Response: Thank you very much for reviewers’ valuable advice, I have deleted the irrelevant information

  1. Several excerpts are only about the theory (literature review) of a particular subject of the study. This needs to be revised.

Response: Thank you very much for reviewers’ valuable advice, I have completed the revision and marked it in red

CONCLUSION

  1. Do not start with “In summary”

Response: Thank you very much for reviewers’ valuable advice, "In summary" has been removed.

Yours Sincerely,

Xingbin Wang

Jie Li

Round 2

Reviewer 2 Report

Comments and Suggestions for Authors

The article has been corrected, according to previous suggestions.

Just a small correction:

In Figure 5, the correct sign is ≤. Please correct.